# Thirteen Independent Genetic Loci Associated with Preserved Processing Speed in a Study of Cognitive Resilience in 330,097 Individuals in the UK Biobank

**DOI:** 10.3390/genes13010122

**Published:** 2022-01-10

**Authors:** Joan Fitzgerald, Laura Fahey, Laurena Holleran, Pilib Ó Broin, Gary Donohoe, Derek W. Morris

**Affiliations:** 1Cognitive Genetics and Cognitive Therapy Group, Centre for Neuroimaging & Cognitive Genomics, School of Psychology and Discipline of Biochemistry, National University of Ireland Galway, H91 TK33 Galway, Ireland; j.fitzgerald24@nuigalway.ie (J.F.); l.fahey5@nuigalway.ie (L.F.); laurena.holleran@nuigalway.ie (L.H.); gary.donohoe@nuigalway.ie (G.D.); 2School of Mathematics, Statistics and Applied Mathematics, National University of Ireland Galway, H91 TK33 Galway, Ireland; pilib.obroin@nuigalway.ie

**Keywords:** cognitive resilience, healthy aging, genomics, processing speed, proxy phenotypes, functional analysis

## Abstract

Cognitive resilience is the ability to withstand the negative effects of stress on cognitive functioning and is important for maintaining quality of life while aging. The UK Biobank does not have measurements of the same cognitive phenotype at distal time points. Therefore, we used education years (EY) as a proxy phenotype for past cognitive performance and current cognitive performance was based on processing speed. This represented an average time span of 40 years between past and current cognitive performance in 330,097 individuals. A confounding factor was that EY is highly polygenic and masked the genetics of resilience. To overcome this, we employed Genomics Structural Equation Modelling (GenomicSEM) to perform a genome-wide association study (GWAS)-by-subtraction using two GWAS, one GWAS of EY and resilience and a second GWAS of EY but not resilience, to generate a GWAS of *Resilience*. Using independent discovery and replication samples, we found 13 independent genetic loci for *Resilience*. Functional analyses showed enrichment in several brain regions and specific cell types. Gene-set analyses implicated the biological process “neuron differentiation”, the cellular component “synaptic part” and the “WNT signalosome”. Mendelian randomisation analysis showed a causative effect of white matter volume on cognitive resilience. These results may contribute to the neurobiological understanding of resilience.

## 1. Introduction

Cognitive decline is one of the most feared aspects of aging leading to major health and social issues and is associated with illness, dementia and death [1]. Non-pathological or age-related cognitive decline leads to increased challenges in completing tasks that require information processing and memory, which in turn leads to a deleterious effect on an individual’s enjoyment of and participation in life events [2]. Cognitive resilience is our ability to withstand negative effects of stress and maintain cognitive functioning. Understanding the factors that contribute to resilience is becoming increasingly important given the aging demographics of the world’s population [3]. There is a growing knowledge of how non-genetic factors such as cardiovascular health and social participation contribute to cognitive resilience [4]; however, an understanding of the genetic contribution has been hampered by the lack of large datasets with genetic data and suitable longitudinal data on cognition. One theory that examines the biological influences on rates of cognitive decline in healthy aging is the concept of reserve, maintenance, and compensation leading to cognitive resilience [5]. Reserve is usually described in terms of both brain reserve, which is the overall strength of size of structural components such as the quantity of neurons and synapses, and cognitive reserve, which refers to adaptability of these components [6]. These are hypothesised to reflect a level of neural resources built up over our lifetime, maintained via the ability to repair cellular damage to maintain cognitive function, with losses compensated for by use of alternative undamaged cognitive functions. In turn, these mechanisms are thought to be mediated by a combination of environmental and genetic factors.

Others propose that variation in the rate of cognitive decline can be explained by variation in intelligence. Longitudinal analysis in the Lothian Birth Cohort has shown that childhood intelligence has a protective effect on cognitive decline in late life [7]. Other studies show that while higher education reflects greater cognitive ability, the rates of change in that ability over time are consistent across all education levels, with those starting at a higher level simply having further to fall before they present with mild cognitive impairment [8,9]. The role of intelligence is confounded by the fact that higher intelligence is associated with healthier life styles, which has a protective effect on cognitive decline [10].

Salthouse proposed the reduced speed of processing hypothesis as earlier studies on cognitive decline showed that processing speed is one of the strongest predictors of performance across cognitive tasks in older adults [11,12]. This theory proposes that older adults take longer to process information and the result of this slower processing leads to impairment in cognitive functions and information is not available for the next part of a task as quickly as in younger adults. It is proposed that superior intelligence is linked to faster processing speed and speed of higher-order information processing explains about 80% of variance in cognitive ability [13]. In a study using 1800 adults ranging in age from 20 to 90, it was found that 70 to 80% of decline in processing speed was shared with declining reasoning ability [14].

The purpose of this study was to explore genetic variation associated with cognitive resilience within the UK Biobank (UKB) [15]. Due to the absence of robust longitudinal data in the UK Biobank, we examined the available cognitive and other phenotypic data to explore cognitive resilience for the first time in a large dataset. After careful consideration and given the growing need to understand cognitive decline in an aging population, we decided on an approach that would make the best use of the available data. The cognitive measure that was most sensitive to age within the UK biobank and that was tested on the largest number of people was reaction time (RT), reflecting an individual’s processing speed, as a measure of current cognitive performance. Processing speed is a key component, and predictor, of cognitive ability [13,16]. In the absence of a direct measure of processing speed at an earlier time point, we used academic achievement measured by number of years in education (education years (EY)) as a proxy phenotype for cognitive performance in early adulthood, following several previous studies [17,18,19]. By separating the population based on years of education, we are assuming that we are capturing individuals that in general have increased processing speed in early life. We created our final phenotype to capture individuals that had preserved their processing speed over a 40 year time span.

Individual differences in processing speed are important in the relationship between executive functioning and academic performance [20]. This approach captures an average time span of 40 years between past and current cognitive performance in the UKB. A confounding factor in this strategy is that EY is highly heritability with a polygenic nature [21] that can mask the genetics of resilience. To overcome this we employed Genomics Structural Equation Modelling (GenomicSEM) [22] to perform a GWAS-by-subtraction [23] using two GWAS, one which captured genetic variants associated with EY and resilience and a second which captured genetic variants associated with EY but not resilience. Subtracting one from the other generated two new GWAS, one capturing EY and the other capturing the genetics of a processing speed-based cognitive resilience phenotype. Replication of this approach was shown using independent discovery and replication samples within the UKB and explored in an independent longitudinal dataset. Full GWAS results were examined further using functional genomics analysis.

## 2. Materials and Methods

### 2.1. Ethics Statement

Our use of UK Biobank data in this study falls within the UK Biobank’s generic Research Tissue Bank (RTB) approval from the NHS North West Research Ethics Committee, UK (reference 11/NW/0382).

### 2.2. The UK Biobank

The UKB is a dataset of over half a million participants between the ages of 40 and 69, recruited from all over the UK in the period of 2006 to 2010 and has been described extensively elsewhere [15]. We obtained permission to access both the phenotypic and genetic data under project # 23739.

### 2.3. Genetic Data

Genotypic data were collected, processed, quality controlled and imputed by the UKB [24]. During our in-house quality control of the imputed data, we excluded samples with a Mahalanobis distance >6 SD from multi-mean of European Population structural analysis, removed samples with discordant sex information, chromosomal aneuploidies, high missingness/heterozygosity, and retracted consent using UKB definitions. Using the UKB-provided file on genomic relatedness, subjects with more than 10 relatives were removed and then one individual from each pair was removed until no related subjects were present. The final sample size used in this analysis was 333,664 participants.

Variants were screened by applying quality control filters (geno 0.02, MAF 0.001, info score 0.09 and HWE 0.0001) and removing duplicates resulted in 8,378,152 variants for use in our final analysis.

### 2.4. Phenotypic Data

Participants undertook a wide range of cognitive tests. The types of tests and the method of collection and reliability are described elsewhere [25,26]. Analysis of cross-sectional cognitive data at time zero using IBM SPSS V24 [27] shows a moderate correlation between age and decline in performance on reaction time and a small correlation with numeric memory, pairs matching, prospective memory, and a weak correlation with fluid intelligence ( Appendix A). Fluid intelligence was repeated at two subsequent intervals; however, no significant sensitivity to aging was found. Deficiencies in the robustness of the longitudinal data collected at the second and third time points have been discussed elsewhere [26]. Follow-up web-based cognitive data were examined as a potential general or ‘g’ factor phenotype; however, this approach had insufficient power to generate meaningful results.

### 2.5. Generation of Resilience Phenotype

Given the lack of longitudinal data, an alternative approach was to use proxy phenotypes. For past cognitive performance, we examined the use of educational attainment/years in education [17,18,19]. Educational attainment is available for 332,089 individuals in the UKB that met our genotypic QC requirements. In the dataset, age completed full time education was recorded for participants who did not go to college but not for those who attended higher education. We therefore assigned a default score of 20 to those who attended college and created a binary phenotype using less than or equal to age 17 to divide participants into two categories—above average and below average education years (EY). We then examined the cognitive data and selected the parameter of processing speed as measured by reaction time (RT) as an indicator of current cognitive performance. RT was chosen as it had a good correlation with age and data were available on most participants (N = 331,495). RT is the speed in milliseconds to correctly identify matching pairs. It was adjusted for age using the slope of the Pearson’s correlation between age and RT and for normality [17], using the natural log of corrected RT. A binary RT variable was created using the natural log mean value (5.71). Those with a value less than or equal to the mean were considered to have faster than average processing speed or RT (quicker to react) and those above the mean were considered to have slower than average processing speed or RT. At total of 330,097 individuals had measurements for EY and RT and genetic data and these made up the final sample (Appendix A).

Using these two binary variables—above or below average EY and faster or slower RT—we created four group of participants (Figure 1b). One of these groups demonstrated high resilience and these were our cases for our first “EY + Res” GWAS who had below average EY previously and faster than average RT now. A second group demonstrated low resilience or cognitive decline, and these were our controls for that GWAS who had above average EY previously and slower than average RT now. The two remaining groups of UKB samples displayed consistent cognitive performance over time. Here, our cases for our second “EY/NonRes” GWAS had below average EY previously and slower than average RT now (below average cognition over time) and our controls had above average EY previously and faster than average RT now (above average cognition over time).

Figure 1 shows an overview of the analysis steps and a detailed description of the process used to generate the resilience variable is included in the methods. Given the multi-step method proposed in this analysis, we sought to confirm findings using our method in an independent sample. Therefore, we divided the UKB into discovery (N = 266,543; 81% of participants) and replication (N = 63,554; 19% of participants) samples (Figure 1a). Sample sizes used for analysis are shown in Appendix A. We used EY as a proxy phenotype measuring past cognitive performance [17,18,19]. Processing speed as measured by RT was chosen as an indicator of current performance given its strong correlation with age and the fact that data were available on most participants in the UKB. We created a binary variable for each measure by using the average score within the dataset to split the participants into similarly sized groups. By combining these two binary variables, we created four groups of participants (Figure 1b). One of these groups demonstrated high resilience and these were our cases for GWAS who had below average EY previously and faster than average RT now. A second group demonstrated low resilience or cognitive decline, and these were our controls for GWAS who had above average EY previously and slower than average RT now. Results for this GWAS were dominated by SNPs associated with EY because the high resilience cases and low resilience controls had below average and above average EY measures, respectively. We named this GWAS “EY + Res” because it identified SNPs associated with both EY and resilience (Figure 1c).

### 2.6. GWAS-by-Subtraction (GBS)

To extract SNPs that were associated with resilience only, we used Genomics Structural Equation Modelling (GenomicSEM) [22]. There are several processing steps that need to be performed to enable the summary statistics to be processed through GenomicSEM and these are described in the original paper by Grotzinger et al. and accompanying tutorials [28]. Closely following the process used by Demange et al. [23], we defined a Cholesky model (Figure 2) as follows using the summary statistics from the EY + Res and EY/NonRes GWASs. Both EY + Res and EY/NonRes were regressed on a latent factor, which captured the shared genetic variance in EY (hereafter “*EduYears*”). EY + Res was further regressed on a second latent factor capturing the variance in EY + Res independent of EY/NonRes, hereafter “*Resilience*”. Genetic variance in *Resilience* was independent of genetic variance in *EduYears* (rg = 0) as the *Resilience* factor represents residual genetic variation in our EY + Res phenotype that is not accounted for by the EduYears factor. These two latent variables, *Resilience* and *EduYears*, were then regressed on each SNP in the original GWASs (EY + Res and EY/NonRes), resulting in new GWAS summary statistics for both *Resilience* and *EduYears* (Figure 2). To calculate the path loadings for λEduYears–EY + Res and λResilience–EY + Res, the model was run without the SNPs. (Refer to https://github.com/joanfitz5/cog.res for Detailed Analysis Steps, accessed on 18 December 2021).

### 2.7. Execution of GBS

To show replication of our GBS-based Resilience GWAS, we divided the UKB into a discovery (81%) and replication (19%) sample. The replication sample included participants in the UKB that had brain imaging data available (N = 37,439) and other random participants to give a total sample size of 63,554. The discovery sample consisted of the remaining suitable participants (N = 266,543) Sample sizes used for analysis are shown in Appendix A. For the discovery, replication and full analysis, we performed two initial GWAS for each sample (EY + Res and EY/NonRes) in plink2.0 [29] using sex, age, assessment centre, genotype array and the first 8 principal components of the population stratification analysis as supplied by the UK Biobank. We then performed GBS on both sets of samples, resulting in a *discovery.Resilience* and *discovery.EduYears* GWAS, a *replication.Resilience* and a *replication.EduYears* GWAS, and later a full *Resilience* GWAS and *EduYears* GWAS.

### 2.8. Calculation of Sample Size after GBS

Running the analysis through GBS alters the sample size and it is necessary to calculate the new value for downstream analysis. To calculate sample size or effective N (Neff) of the *Resilience* GWAS for test, replication and full, we followed the procedure specified in GenomicSEM [30] and by Demange et al. (see Appendix B URLs). To do this, we needed to determine path loading for the models used in the three analyses as the path loading differs with different sample sizes. We trimmed our data to only include SNPs with a MAF of >0.10 and <0.40, as low and high MAF can bias the result. The output of this analysis and the calculations of sample size are in Appendix A.

### 2.9. Additional GWAS

To examine the effect of GBS on EY + Res, we ran two further case–control GWAS of above/below average EY and faster/slower than average age-corrected RT. We mirrored the sample size used to generate EY + Res and EY/NonRes by randomly selecting 82,000 samples as cases and controls from the dataset. This analysis was run in plink 2.0 [29] using sex, age, assessment centre, genotype array and the first 8 principal components of the population stratification analysis as supplied by the UKB. To further interpret the relationship of resilience to RT we performed a quantitative GWAS on the 164,000 samples used in the cases control study. In addition, we also performed a full dichotomised study of all 333,664 individuals in the UK biobank with RT data.

### 2.10. Identification of Genomic Loci Associated with Resilience

Manhattan plots of GWAS outputs from original phenotypes and GBS outputs were generated in FUMA v 1.3.6 [31] using a *p*-value setting of <5 × 10^−8^ for genome-wide significant SNPs. We used an LD *r*^2^ setting of 0.6 and the 1000 G phase 3 European reference panel to identify independent lead SNPs and an additional *r*^2^ setting of 0.1 to identify lead SNPs and a maximum distance for LD blocks of 250 kb to separate findings into separate genetic loci. Conditional analysis was performed where there was more than one independent significant SNP within 1000 kb distance using the “--condition” command in Plink 1.9 [29], which adds a SNP as a covariate in GWAS analysis.

FINEMAP [32] was used to investigate causal SNPs by analysing the relationship between the candidate GWAS SNPs generated in FUMA and LD data. LD files were generated in plink 1.9 using the “--r square spaces” command. Results of SNPs listed by Bayes Factor for each locus were examined as well as the configuration files generated by FINEMAP to examine for causal SNPs sets. The maximum number of SNPs in a set was fixed at 3.

### 2.11. Function Analysis of GWAS Output

We used FUMA v 1.3.6 [31] to perform functional analysis. We used the default settings as described in the Tutorial section of the website and in previous publications [33,34]. FUMA analysis of *Resilience* is published and can be viewed publicly in FUMA as ID:171. We used the calculated effective sample size of 111,316 (Neff) for the analysis of the *Resilience* output to examine the functional consequences of SNPs on genes, Combined Annotation-Dependent Depletion (CADD) scores, chromatin states and Regulome DB analysis.

### 2.12. Mapping SNPs to Genes

Gene mapping was performed in FUMA using three strategies: (a) *positional mapping* which mapped SNPs to genes based on their genomic location within a 10 kb window of known gene boundaries. (b) *Expression quantitative trait (eQTL) mapping* which aligned cis-eQTL SNPs to genes whose expression they affected, selecting information from tissue types in 4 datasets in FUMA (PsychENCODE [35], BIOS QTL [36], Blood eQTL [37], and GTEx 8 [38]). (c) *Chromatin interaction mapping* using the 3D DNA to DNA interactions mapped SNPs to genes.

Gene-set analyses: The GENE2FUNC function within FUMA examines enrichment of mapped genes using hypergeometic tests of 9494 gene sets from GTEx [39], MSigDB [40] and GWAS catalog [41].

### 2.13. MAGMA Gene-Based Analysis

FUMA computes a gene-based genome-wide association analysis (GWGAS) from the SNP-based *p*-value from the GWAS. A total of 18,879 protein-coding genes containing a minimum of one GWAS SNP were used in this analysis and were used to test for association with 53 tissue types. Associations were Bonferroni corrected for multiple testing with a *p* value threshold of <0.05/18,879 = 2.648 × 10^−6^.

We further explored the sets of associated genes in cell type specificity analyses with scRNA-seq in FUMA [42] using the following datasets: GSE104276 human prefrontal cortex per ages [43], GSE67835 human cortex [44] and Linnarsson Mouse Brain Atlas [45]. We analysed significant cell types across datasets, independent cell type associations based on within-dataset conditional analyses and pair-wise cross-datasets conditional analyses.

### 2.14. Comparison with Published Traits

LD score regression (LDSR) analysis was performed using the LDSC function within GenomicSEM [22] to examine the genetic correlation between Resilience with other phenotypes. Various sources were used to obtain summary statistics from GWAS of published research in psychiatry, brain imaging, and other traits of interest (Appendix A). Munged summary statistic files generated during GBS were used for *Resilience*, *EduYears*, EY + Res and EY/NonRes in the LDSR. Associations were Bonferroni corrected for multiple testing with a *p* value threshold of 0.05/21 = 2.88 × 10^−3^.

### 2.15. Mendelian Randomisation

Mendelian randomisation was performed using Generalised Summary statistics-based Mendelian Randomisation [46] (GSMR) using the GCTA tool. The procedure examines credible causal associations between different traits based on GWAS outputs and requires non-overlapping samples. This restricted our analysis because most of the traits examined by LDSC contained UKB participants. However, the sample used for the *discovery.Resilience* GWAS (Section 2.7) does not contain individuals that have imaging data within the UKB, so we used this cohort to examine unidirectional and bidirectional causal associations between *Resilience* and phenotypes that showed significant correlations with *Resilience* using LDSC. We used a HEIDI-outlier *p*-value of 0.01 for outlier detection analysis. Given the low level of independent significant SNPs in the *discovery.Resilience* GWAS and the imaging GWAS, we reduced the default minimum level of significant SNPs from 10 to 8. For the disorders of ALS, bipolar disorder and schizophrenia, we used the full Resilience GWAS and ran the analysis at the default setting of a minimum of 10. Associations were Bonferroni corrected for multiple testing with a *p* value threshold of 0.05/12 = 4.23 × 10^−3^.

### 2.16. US Health and Retirement Study (HRS)

The HRS is a longitudinal study of adults aged 50 years or older in households in the United States. The study commenced in 1992 and participants were interviewed at baseline and every two subsequent years. Phenotypic data were accessed through the HRS web site (see Appendix B URLs). Genetic data were supplied through dbGaP as project 18937.

### 2.17. Genetic Data

The imputed genetic data were available per chromosome as probability files (gprob.gz). Pgen files were created and were screened by applying quality control filters (geno 0.02, MAF 0.01, info score 0.09 and HWE 0.000001) and removing duplicates for each chromosome. In addition, SNPs in the kgp format were converted to RSID using the USCS browser [47].

### 2.18. Cognitive Phenotypes

Several cognitive tests were administered by interviewers, by phone or face to face. These tests are described in detail on the HRS website. The HRS dataset does not contain a reaction time or processing speed measure. To maximise the number of participants with two data points, we consisted of the interviewer reading a randomised list of 10 nouns to the respondent from one of four lists, and afterwards asked the respondent to recall as many words as possible.

Data were available for eleven intervals, each two years apart, representing a time span of 22 years. Cognitive data were available for 38,183 participants. Of these, 15,620 had genetic data and this was reduced further when individuals were eliminated based on ethnicity (non-Caucasian), missing data or less than 2 data points, to reduce the available sample to 9526 individuals. Linear mixed modelling (LMM) in SPSS was used to analyse the data as this method allows for missing data in longitudinal samples, using the covariates of gender, birthyear, education, time of test and quadratic time point [48].

### 2.19. GWAS of Cognitive Change over Time

We created a cognitive phenotype for cognitive change from the LMM output and performed a GWAS using the first six PCA components as covariates as recommended in the genotype QC report [49]. The output of the GWAS was processes through FUMA.

## 3. Results

### 3.1. Initial Phenotype Development

In order to identify SNPs that were associated with resilience alone and remove SNPs that were associated with the EY component of the phenotype, we performed a second GWAS using the two remaining groups of UKB samples that displayed consistent (i.e., unchanging) performance over time. The first of these groups consisted of those with below average EY previously and slower than average RT now (i.e., consistently below average performance over time); the second group consisted of those who showed above average EY previously and faster than average RT now (i.e., consistently above average performance over time). We named this GWAS “EY/NonRes” because it identified SNPs associated with EY but not resilience (Figure 1d).

We then used GWAS-by-subtraction (GBS) [23] to subtract the results of EY/NonRes from EY + Res to leave SNP associations with resilience. This method uses GenomicSEM [22] to integrate GWAS into structural equation modelling (Figure 2). Both EY + Res and EY/NonRes were regressed on a latent factor, which captured the shared genetic variance in EY, hereafter “*EduYears”*. EY + Res was further regressed on a second latent factor capturing the variance in EY + Res independent of EY/NonRes, hereafter “*Resilience*”. Genetic variance in *Resilience* was independent of genetic variance in *EduYears* (*r_g_* = 0) as the *Resilience* factor represents residual genetic variation in our EY + Res phenotype that is not accounted for by the *EduYears* factor. These two latent variables, *Resilience* and *EduYears*, were then regressed on each SNP in the original GWASs (EY + Res and EY/NonRes), resulting in new GWAS summary statistics for both *Resilience* and *EduYears* (Figure 1e).

### 3.2. Discovery and Replication Analysis within the UKB

For the discovery sample, we performed the two initial GWASs (discovery.EY + Res and discovery.EY/NonRes) and then performed GBS on both sets of samples, resulting in *discovery.Resilience* GWAS results and *discovery.EduYears* GWAS results. We repeated this for the replication sample to produce *replication.Resilience* GWAS results and *replication.EduYears* GWAS results. Comparison of the *discovery.Resilience* GWAS with the *replication.Resilience* GWAS by LD score regression (LDSR) analysis [22] showed extremely high correlation between the two datasets (*r*_g_ = 0.964, *p* = 4.45 × 10^−44^). The *discovery*.*Resilience* GWAS was then processed through FUMA v 1.3.6 [31] and ten independent genome-wide significant SNPs were identified. When compared to the *replication*.*Resilience* GWAS, there was a consistent direction of effect for all ten SNPs (Binomial sign test, *p* = 9.77 × 10^−4^). Five of the ten SNPs were significant after Bonferroni multiple test correction for those SNPs tested (*p* < 0.005). Thus, we demonstrated that we could replicate genetic associations with *Resilience* in an independent sample. Results for the ten independent genome-wide significant SNPs and their replication analysis are in Appendix A.

### 3.3. Analysis of the Full Sample

Next, in order to increase the statistical power of our analysis, we combined both the discovery and replication samples to run an analysis on the full sample (N = 330,097). This resulted in initial EY + Res and EY/NonRes GWASs and following GBS, *Resilience* GWAS results and *EduYears* GWAS results. SNP-based heritability estimate analysis showed a h^2^ value of 0.13 (SE = 0.006) for *Resilience*. For comparison in similarly sized samples, we also ran GWASs of EY and RT using participants randomly selected from the UKB (EY, N = 82,000 above average EY cases and N = 81,999 below average EY controls; RT, N = 82,000 faster than average RT cases and N = 82,000 slower than average RT controls). These comparisons are shown in Appendix A. A Manhattan plot and a quantile–quantile (Q–Q) plot of *Resilience* on the full sample is shown in Figure 3a,b. Manhattan plots for the other five GWAS (EY + Res, EY/NonRes, *EduYears*, EY and RT) are in Appendix A.

Initially, both EY + Res and EY/NonRes had a strong negative correlation with EY (*r*_g_ = −0.88 and *r*_g_ = −0.89, respectively (Appendix A). The strength of these correlations likely reflects the major contribution of EY to these phenotypes and they are negative because for EY + Res and EY/NonRes, the direction of effect is in the opposite direction to EY, as the cases are low EY, whereas for the EY GWAS, the cases are high EY. EY + Res and EY/NonRes had a moderate positive correlation with each other (*r*_g_ = 0.54). After GBS, there was no genetic correlation between *Resilience* and *EduYears* (*r*_g_ = 0.01, *p* = 0.803), suggesting that the subtraction had successfully separated out the genetic associations for both phenotypes.

Although the EY component of *Resilience* was addressed by the GBS method, the RT component was not and the genetic correlation between *Resilience* and RT was strong (*r*_g_ = 0.80; Appendix A). This finding was examined further following functional analysis of associated loci with detail on this provided at the end of Results.

### 3.4. Analysis in a Dataset Independent of the UKB

The US Health and Retirement study (HRS) is a longitudinal study of adults aged 50 years or older in households in the United States [49]. The study commenced in 1992 and participants were interviewed at baseline and every two subsequent years. In order to maximise sample size, we used immediate memory recall as our cognitive measure. Using linear effect modelling [48] we generated a cognitive change phenotype and ran a GWAS using available participants (N = 9526). Although no significant genetic loci were found in this analysis and only one of our *Resilience* genetic loci was nominally significant in the HRS GWAS (chromosome 3 locus; *p* = 0.025), there was a significant genetic correlation (*r*_g_ = −0.65, *p* = 1.5 × 10^−3^) between cognitive change and *Resilience*. This negative correlation is expected as the HRS GWAS used a measure of cognitive change, whereas the *Resilience* GWAS used a measure of resilience to change. Manhattan and Q–Q plots of the HRS cognitive change GWAS are in Appendix A.

### 3.5. Functional Analysis

#### 3.5.1. Description of Genetic Loci

Function analysis was performed on *Resilience* in FUMA v 1.3.6 [31].

A total of 1329 significant SNPs were tagged from the *Resilience* GWAS and were associated with 26 independent lead SNPs (*p* < 5 × 10^−8^). Including SNPs in the reference panel that are in LD with the independent SNPs resulted in a total of 1922 candidate SNPs. Functional annotation of the candidate SNPs showed that 82% were intergenic/intronic. A total of 84 SNPs had a Combined Annotation-Dependent Depletion (CADD) score greater than the threshold of 12.37 which indicates that the variants are potentially pathogenic [50] (see Appendix A).

Lead SNPs were grouped into 13 independent genetic loci that are on 9 different chromosomes. Detailed maps of each locus are available in Supplementary Figure 3. Conditional analyses showed that the significance of all independent lead SNPs at each locus was reduced when the GWAS was conditioned for the index or most associated SNP, confirming the linkage of the index SNP to each lead SNP (Appendix A).

#### 3.5.2. Fine Mapping

FINEMAP [32] was used to provide further information on significant SNPs in LD with the index SNP on each locus using the GWAS SNPs generated by FUMA (Appendix A). The log_10_ Bayes Factor (B_10_) quantifies causal evidence for a particular SNP and a posterior probability value yielding a B_10_ greater than 2 indicates considerable evidence of causality [32]. One SNP, rs62074125, on chromosome 17, exceeded this value (B_10_ = 2.64). This SNP is an intron within the *WNT3* gene, which is associated with cognitive function [51]. The next highest result was on chromosome 4 where rs2189234 had a value slightly below 2 (B_10_ = 1.62). This SNP is an intronic variant in the *TET2* gene, which is discussed below. FINEMAP analysis showed that the index SNP had the highest Bayes Factor for all loci with four exceptions (Appendix A).

#### 3.5.3. Gene Mapping

Three approaches were used in FUMA to map the associated variants to genes: (a) positional mapping mapped 141 SNPs to genes based on their genomic location within a 10 kilobase window of known gene boundaries. (b) Expression quantitative trait (eQTL) analysis mapped 207 cis-eQTL SNPs to genes whose expression they were associated with. (c) Chromatin interaction analysis using the 3D DNA to DNA interactions mapped SNPs to 243 genes. Circos plots for all loci are included in Appendix A. The circos plot from chromosome 3 shows that 102 genes were mapped to this region, representing 42% of the total genes mapped. In addition, the circos plot from chromosome 17 shows two distinct clusters of SNPs. Genes in this region (*MAPT*, *WNT3*, *CRHR1*, *KANSL1*, and *NSF*) have been previously associated with general cognitive function but also with other cognitive indicators [51]. Details of this gene mapping analysis is in Appendix A.

In addition to the three approaches above, we also performed a genome-wide gene-based association analysis (GWGAS) using the MAGMA function within FUMA [31], which looks at the aggregate association results of all SNPs in a gene in contrast to the previous analyses that examined the association signals at the level of individual SNPs. A GWGAS was performed using the *Resilience* GWAS on 18,879 protein-coding genes containing at least one SNP from the GWAS. Based on the number of genes tested, a Bonferroni-corrected threshold of *p* < 2.65 × 10^−6^ was used (see Q–Q plot of this association—Figure 3c). A total of 52 protein-coding genes were identified as associated, 40 of which were identified by the previously described strategies (Appendix A). In total, 33 genes were identified by all four mapping strategies (Figure 3c and Appendix A).

Many of these 33 genes have been connected with cognitive performance, neurodegenerative disorders or aging and represent potential therapeutic targets: *STAU1* (chr 20) and *SEMA3F* (chr 3) are predicted to control cognitive decline in aging through formation of neural circuits and synaptic transmission [52]. *BNS* (chr 3) codes for bassoon presynaptic cytomatrix protein which is implicated in the regulation of neurotransmitters at inhibitory and excitatory synapses [53]. *IP6K1* (chr 3) codes for inositol pyrophosphate biosynthesis, and mouse studies have shown its involvement in short-term memory by altering presynaptic vesicle release and short-term facilitation of glutamatergic synapses in the hippocampus [54]. *MST1* (chr 3) has been shown to play a role in protecting cells from oxidative stress which leads to aging and eventual cell death [55]. *TET2* (chr 4) codes for ten eleven translocation methyl cytosine dioxygenase 2 which catalyses the production of 5-hydroxymethylcytosine and is associated with increased neurogenesis in the hippocampus and cognition in animal studies [56]. *ATXN2* (chr 20) is involved in regulating mRNA and is linked to decline in cognitive function in older adults [57], general cognitive function [51] and neurodegenerative disorders [58]. The *ATXN2/BRAP* locus has a strong association with parenteral lifespan [59]. Another mapped gene close to *ATXN2* and *BRAP* is *SH2B3*, which encodes lymphocyte adaptor protein LNK, and plays a role in human aging though the mechanism involved is not fully understood [60]. The gene *ALDH2* (chr 12) codes for aldehyde dehydrogenase and there is a link between this enzyme and life span as well as cardiovascular aging [61].

Among the associated SNPs at the 33 prioritised genes are two UTR3 variants on chromosome 3 (rs2681781 (CADD = 17.77) and rs4625 (CADD = 15.6)) that map to *RBM5* and *DAG1*, respectively. Animal studies have shown that *RBM5* is a likely regulator of Rab4a, which in involved in many neurobiological functions including the transport of transmembrane proteins required for neurotransmission [62]. *DAG1* has been associated with increased cognitive performance and is associated with GABAergic signalling in the hippocampus [63]. In addition, one other variant of note is rs1130146 that maps to *DDX27* (chr 20), a gene that was mapped by all strategies except for GWGAS and is associated with longevity [64]. This missense SNP has a CADD score of 31 and is predicted by SIFT to be deleterious and by PolyPhen to be possibly damaging.

#### 3.5.4. Tissue, Cell Type and Pathway Enrichment Analysis

Using gene expression data for 53 tissues obtained from GTEx [65], we found all brain regions to be significantly enriched for our associated genes with the strongest enrichments for the frontal cortex, BA9 (*p* = 2.26 × 10^−11^), the cortex (*p* = 8.48 × 10^−11^) and the cerebellar hemisphere (*p* = 1.18 × 10^−10^; Figure 4a and Appendix A). There was no significant enrichment in other tissues of the body. Expression analysis at the cellular level was performed using datasets from the human prefrontal cortex by age [43], the human cortex [44] and Linnarsson Mouse Brain Atlas [45]. We analysed significant cell types across datasets, independent cell type associations based on within-dataset conditional analyses and pair-wise cross-datasets conditional analyses (Figure 4b and Appendix A). These analyses identified four neuronal cell types to be enriched for our associated genes. For human data, these were neurons in the cortex (*p* = 2.16 × 10^−6^), and GW26 GABAergic neurons in the prefrontal cortex (*p* = 6.59 × 10^−^^8^). For mouse data, these were excitatory glutamatergic neurons in cortical pyramidal layer 5 of the cerebral cortex (TEGLU10; *p* = 6.98 × 10^−^^6^) and excitatory glutamatergic/nitric oxide neurons in the tegmental reticular nucleus of the pons in the hindbrain (HBGLU8; *p* = 6.74 × 10^−7^). The enrichment in GABAergic neurons is interesting because there is growing evidence to suggest that impairment of the GABAergic system caused by aging results in an imbalance in the inhibitory/excitatory process involved in the neuronal response to cellular challenges and environmental changes. This results in increased vulnerability to synaptopathy and cognitive decline [66].

Gene-set analysis performed on curated gene sets and Gene Ontology (GO) [67] terms using the full distribution of SNP *p*-values from the *Resilience* GWAS identified two GO terms to be significantly enriched after adjustment for multiple testing These were the biological processes“ neuron differentiation” (*p* = 9.7 × 10^−7^) and the cellular component “synaptic part” (*p* = 2.14 × 10^−6^). Bi-directional conditional analysis using MAGMA 1.08 [68] showed that these two annotations were independent of each other (Appendix A). One term that was nominally significant for enrichment with a very large β value was “Wnt signalosome” (*β* = 1.22, *p* = 4.75 × 10^−6^). There are 12 genes in this gene set and 6 of the 12 genes had nominally significant *p* values and three of these remained significant after correcting for multiple testing (*p <* 0.0042) (Appendix A). Two of these genes are mapped in *Resilience* loci by eQTL analysis. The first is *APC* on locus 5A. The other is *WNT3* on locus 17 (Appendix A). Deficient Wnt signalling is associated with loss of cognitive ability [69].

#### 3.5.5. Genetic Correlations with Other Traits

We compared our *Resilience* GWAS with recent published GWAS of cognitive phenotypes, psychiatric and neurological disorders, and global brain imaging phenotypes using LDSR analysis. A moderate negative correlation of *Resilience* with intelligence [34] (*r_g_*= −0.26, *p* = 1.29 × 10^−17^ and educational attainment [21] (*r_g_*= −0.45, *p* = 1.64 × 10^−56^ is as expected given that the resilience phenotype was derived from individuals within the UK biobank that has lower than average education years. Of the 13 independent genome-wide significant SNPs for *Resilience*, 6 are associated with intelligence at genome-wide significant levels (*p* < 5 × 10^−8^) but the remaining 7 SNPs are not associated with intelligence (*p* > 0.01). This indicates that some of genetic basis of *Resilience* does not overlap fully with the genetics of intelligence.

When genetic correlation analyses between *Resilience* and psychiatric phenotypes were corrected for multiple testing (P_bon_ < 2.4 × 10^−3^), *Resilience* had a small positive correlation with unipolar depression [70] (*r*_g_ = 0.17, *p* = 5.0 × 10^−10^), a small negative correlation with schizophrenia [71] (*r*_g_ = −0.18, *p* = 1.24 × 10^−12^) and bipolar disorder [72] (*r_g_* = −0.17, *p* = 1.84 × 10^−7^), and a nominally significant negative correlation with neuroticism [73] (*r*_g_= −0.07, *p* = 2.02 × 10^−2^). Examination of neurological disorders showed *Resilience* had a small nominally significant correlation with amyotrophic lateral sclerosis (ALS) [74] (*r*_g_ = −0.21, *p* = 1.44 × 10^−2^), stroke [54] (*r*_g_ = 0.08, *p* = 1.89 × 10^−2^), and Parkinson’s disease [75] (*r*_g_ = −0.08, *p* = 4.58 × 10^−2^), but Alzheimer’s disease (AD) [33] was not significant (*r*_g_ = 0.04, *p* = 0.358) (Appendix A and Figure 5). The lack of correlation with Alzheimer’s disease may be due to the difference in the genetic profile of both phenotypes. It has been shown that decline seen in Alzheimer’s disease (AD) is not an acceleration of the healthy ageing process but has a unique pathology of its own [76].

The GWAS of 11 brain phenotypes from the UK Biobank [77] were examined by LDSC for genetic correlation with the *Resilience* (Appendix A and Figure 5). The volume of global white and grey matter and cerebral white matter in the left and right hemisphere were examined based on the relationship between brain volumes and cognition [78]. Volume of cerebrospinal fluid was included based of its documented association with brain atrophy [79] and the hippocampus, amygdala and nucleus accumbens were examined as moderators of cognitive function [80,81]. After adjusting for multiple testing (P_bon_ = 2.4 × 10^−3^), the only significant correlations found were for white matter volumes where a small positive correlation was found between *Resilience* and global white matter volume (*r*_g_ = 0.14, *p* = 1.19 × 10^−3^), and the volume of cerebral white matter in the left (*r*_g_ = 0.148, *p* = 1.74 × 10^−3^) and right hemisphere (*r*_g_ = 0.160, *p* = 7.34 × 10^−4^).

The correlations of cognitive and psychiatric and neurological disorders are largely supported by gene enrichment analysis of the genes associated with *Resilience* here and previous GWAS of cognitive and psychiatric phenotypes. An analysis of published research from the GWAS catalog [41] showed that the significant SNPs found in this study were previously cited 294 times. A total of 47% of these citations were from studies of cognitive phenotypes (educational attainment, cognitive ability, maths ability and RT) and 5% were from studies of psychiatric disorders (Appendix A). In addition, when this exercise was repeated for overlapping mapped genes, we found that there was considerable overlap with these phenotypes amongst others. The most significant overlap was where 40 mapped genes in the *Resilience* analysis overlapped with the 99 reported genes for short sleep duration (*p* = 2.03 × 10^−57^). In a recent Mendelian randomisation study on sleep duration, it was suggested that sleep duration may represent a potential causal pathway for differences in cognitive ability [82] and increase sleep in adults over 60 is associated with poorer cognitive function [83]. There was also a significant overlap with genes associated with extremely high intelligence [84] where 32 *Resilience* mapped genes overlapped with the 81 associated genes reported in that study (*p* = 1.17 × 10^−45^). Many of the overlapping genes for sleep duration and extremely high intelligence were on chromosome 3 (Appendix A).

#### 3.5.6. Mendelian Randomisation

To investigate whether genetic correlations reflected directional effects, we examined the potential credible causality of the relationship between *Resilience* and phenotypes where independent samples were available using Generalised Summary statistics-based Mendelian Randomisation [46] (GSMR) (Appendix A). We observed a significant bidirectional causal effect of *Resilience* on schizophrenia (*bxy* = −0.25, *p* = 7.02 × 10^−9^) and schizophrenia on *Resilience* (*bxy* = −0.07, *p* = 3.80 × 10^−7^) indicating an inter-relationship between the two phenotypes. By contrast, bipolar disorder and ALS did not have significant credible causality relationships with *Resilience*.

GSMR analysis was also performed using white matter volume variables and *Resilience.* To maintain independence between GWAS datasets, we used the *discovery*.*Resilience* GWAS that did not include UKB participants with imaging data. The low level of independent significant SNPs in the discovery GWAS did not allow for analysis of the causal effect of *Resilience* on white matter. A nominally significant causal association of white matter volume with *Resilience* was detected (*b_xy_* = 0.13, *p* = 0.049) along with causal associations of left and right cerebral hemisphere white matter volume with *Resilience*. The association with the right hemisphere survived multiple test correction (left: *b_xy_* = 0.15, *p* = 0.005; right: *b_xy_* = 0.17, *p* = 0.002). There is no evidence of substantial pleiotropy in the GSMR analysis.

#### 3.5.7. Examination of the Relationship of Resilience with RT

Given the strong positive correlation of *Resilience* with RT (*r*_g_ = 0.80), a possible concern was that we were just identifying genetic associations with RT that are independent of EY. To examine this further we performed a functional analysis on a GWAS of a dichotomised RT phenotype using all suitable participants in the UK Biobank (N = 333,664). This GWAS was perfectly correlated (*r*_g_ = 1, *p* = 7.24 × 10^−115^) with a previously published GWAS where RT was studied as a quantitative phenotype [51]. We found that while nine of the 13 loci identified in the *Resilience* GWAS overlapped with RT, four loci did not (Appendix A). There was a total of 534 mapped genes for RT and 366 for *Resilience*. Of these, 301 were unique to RT and 133 unique to *Resilience* with 223 shared genes. Only 11 of our 33 prioritised genes in *Resilience* were among the 27 prioritised genes for RT (Appendix A). Pathway enrichment analysis identified GO terms that were enriched for both RT and *Resilience* associated genes (e.g., the cellular component “synaptic part”) but also showed pathways related to neuronal processes that are only significant in *Resilience* (e.g., the biological process “neuron_differentiation”; Appendix A). The genetic correlation between *Resilience* and RT is strong because this is an RT-based resilience phenotype. However, there are differences in the associated genes being detected and prioritised because we detected SNPs associated with faster than average RT in individuals that previously showed below average EY, i.e., the resilience phenotype in this study.

#### 3.5.8. Effect of Large Locus on Chromosome 3

Almost 30% of the mapped genes and over 50% of prioritised genes can be attributed to a single large locus on chromosome 3: 49385350–50250837. We were concerned that this locus might have an inflated influence on the functional analysis of *Resilience*. To investigate this, we extracted all SNPs in this locus from the GWAS of *Resilience* and reprocessed the GWAS through FUMA. This analysis showed a decrease in candidate SNPs and mapped genes; however, the main findings of the functional analysis remained unchanged.

## 4. Discussion

This is the first study, to our knowledge, to explore the genetic basis of cognitive resilience in a large dataset, here using processing speed measured in later adulthood as a basis for a resilience phenotype. In the absence of longitudinal data, we used a proxy phenotype of EY to measure cognitive performance in earlier adulthood and have combined case–control GWAS with structural equation modelling to extract genetic variants associated with *Resilience* in the UKB. We have shown the robustness of this method by confirming associations detected in a discovery sample replicate in an independent sample. We have successfully identified 13 independent genome-wide significant loci, resulting in 366 mapped genes and 33 prioritised genes for *Resilience*. Functional analysis showed significant expression of associated genes in all brain tissues, and particularly in the frontal cortex. Significant enrichment of associated genes was also found at the cellular level in both GABAergic and glutamatergic neurons indicating an excitatory/inhibitory control in the prefrontal cortex, and within biological processes related to neuron differentiation, synaptic activity, and WNT signalling.

Mapping of GWAS results identified genes that have been previously associated with cognitive decline including *STAU1*, *SEMF3A*, *IP6K1*, *MST1*, the *ATNX2/BRAP* locus, *ALDH2* and *DDX27*, where a likely functional missense variant is highly associated. Other associated genes involved with synaptic activity and neurogenesis include *BNS*, *DAG1*, *IP6K1* and *TET2*, pointing to potential targets for improvement of cognitive resilience.

A limitation of our study was our reliance on RT to create the *Resilience* phenotype, which results in a strong genetic correlation between *Resilience* with RT. This reflected our study design that detected SNPs associated with faster than average RT or processing speed in individuals that previously showed below average EY. However, the majority of genes prioritised by our *Resilience* GWAS are not prioritised by the RT GWAS and vice versa. We conclude that these findings point to genes that enhance maintenance of processing speed over the life span. Decline in processing speed is a strong predictor of decline in cognitive processing in older adults [11] and had been found to be associated with cerebral small vessel disease and factors involved in the maintenance of cerebellar morphology [85]. In addition, better cognitive processing speed is also associated with larger cerebral cortex volumes (supporting our finding of a causal relationship with white matter volume), lower levels of inflammatory markers and insulin and is mediated by physical exercise [86]. Over half of our genome-wide significant loci for *Resilience* are not associated with intelligence, indicating that factors such as reserve, compensation and maintenance may play a role over and above overall intelligence in determining resilience.

Our use of a proxy phenotype for past cognitive performance and GBS to generate a *Resilience* GWAS was in response to the limitation of not having direct repeated measurements of same cognitive phenotype in large numbers of genotyped samples over an extended time period. Instead, we refined our analysis to make best use of available phenotypic data in the UKB to exploit the large sample size available for genetic discovery. We used the proxy phenotype of academic achievement (EY) to represent past cognitive performance in the absence of a direct measure of processing speed. EY itself is not a predictor of cognitive decline [87]; however, it is associated with cognitive performance in younger adults [9]. Our assumption is that superior processing speed is a necessary component of academic success. Processing speed is indeed the key predictor of number sense, fluid intelligence and working memory, which in turn predict individual difference in academic achievement [88]. Studies using EEG have shown that higher processing speed can explain about 80% of variance in general intelligence. A more efficient transmission of information between frontal attention to memory storage and retrieval benefits those with higher intelligence [20]. The non-cognitive contributors to academic achievement also affect this proxy phenotype [23]. There is debate within the cognitive scientific community as to the suitability of education achievement as a proxy for cognition, some showing a modest correlation [89] while others describe a high phenotypic and genetic correlation [18].

A further limitation is that processing speed as measured by RT is only one component of cognition and it may not be possible to extrapolate the results of this analysis to global cognitive resilience. To support our approach, we explored the use of a much smaller dataset (HRS) which had longitudinal data over time. We showed a genetic correlation between our *Resilience* findings in the UKB and cognitive change in HRS. This dataset was not sufficiently powered to confirm the *Resilience* genetic variants. However, this can be addressed by various biobanks that plan new data collection in the future.

This study demonstrated a new method to explore cognitive resilience and identifies associated loci and genes that may provide neurobiological insights for this processing speed-based resilience phenotype. Given the limitations in the method used to construct the resilience phenotype, these findings are preliminary and will need to be confirmed by future research; however, the findings suggests that cognitive resilience is not just a function of superior intelligence and is causally related to variation in white matter volume. This is turn may represent a potential target for studies seeking to enhance resilience therapeutically.

## Figures and Tables

**Figure 1 genes-13-00122-f001:**
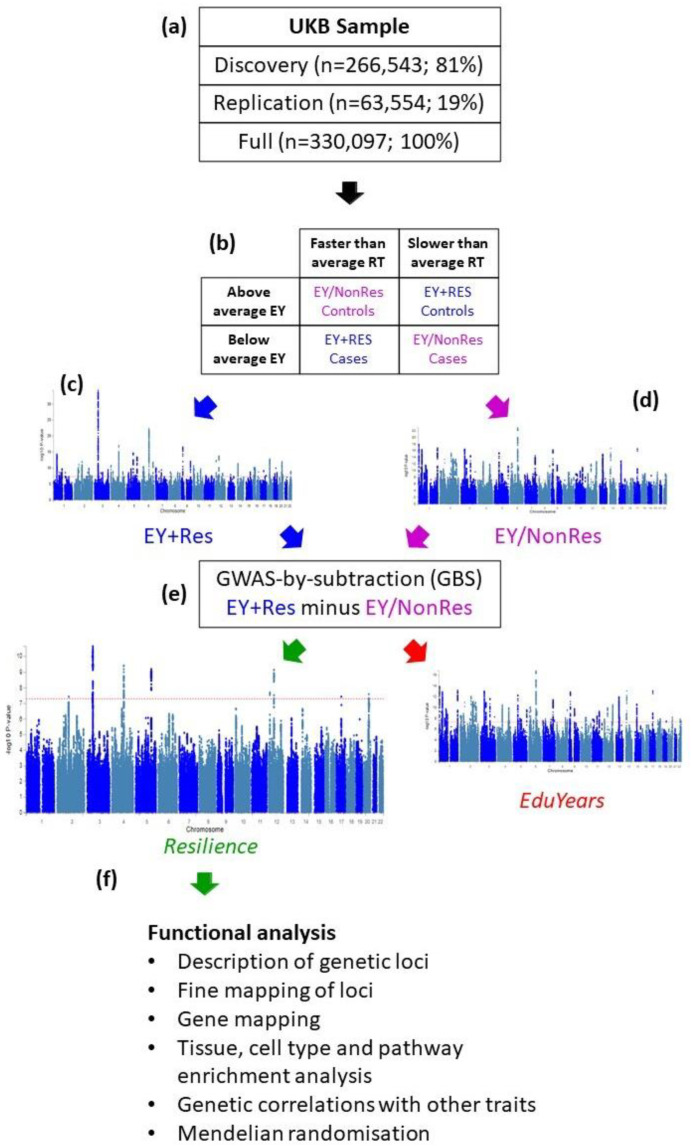
Flow chart of study design. (**a**) The available UKB samples were split into Discovery (81%) and Replication (19%) samples. Following successful replication analysis, the Full sample was also put through the analysis pipeline. (**b**) For Discovery, Replication or Full, samples were assigned to one of four categories based on their EY and RT measures. (**c**) EY+Res cases and controls were analysed in a GWAS. (**d**) EY/NonRes cases and controls were analysed in a GWAS. (**e**) GBS used to subtract the genetic signals for EY/NonRes from EY+Res to result in a Resilience GWAS and an EduYears GWAS. (**f**) Resilience GWAS functionally analysed to identify associated SNPs and genes, and enriched tissues, cell types and pathways, identify genetic correlations with other traits and explore causal relationships between resilience and other traits using Mendelian randomisation.

**Figure 2 genes-13-00122-f002:**
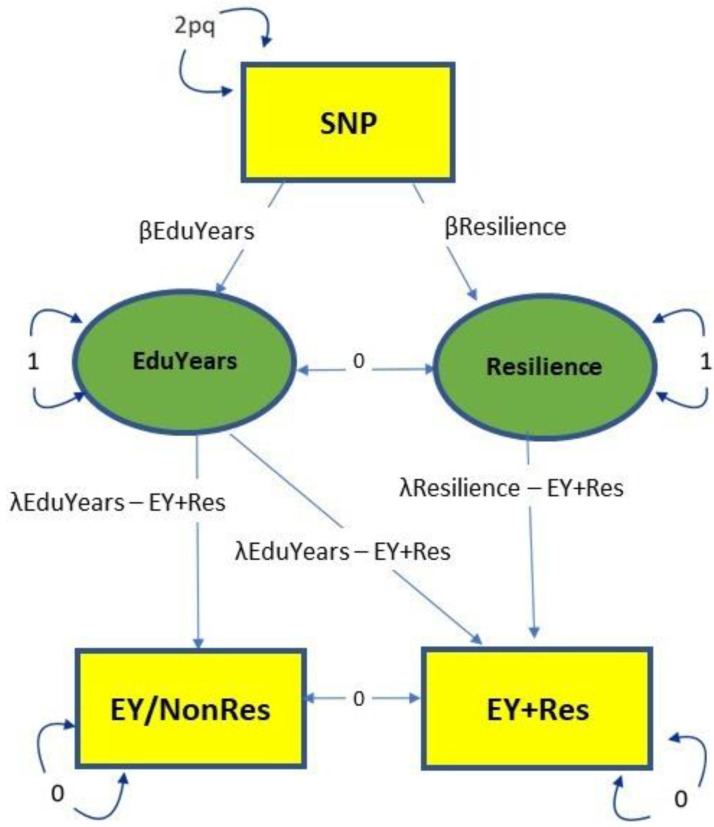
SEM of GWAS-by subtraction model. The observed variables are the GWAS EY + Res and EY/NonRes and SNP and the latent variables (unknown) are Resilience and EduYears. There are two pathways for the SNPs analysis in this model to EY + Res—the first is through EduYears to EY + Res and EY/NonRes and incorporates the genetic effects of the variables used in the phenotype. The other path is through Resilience to EY + Res and measures the genetic effect of resilience independent of EduYears. To calculate the model, the genetic covariances between EY + Res and EY/NonRes and Resilience and EduYears are set to 0 and the variances of EY + Res and EY/NonRes are also set to 0. The variance is therefore explained by the latent factors. The SN*p* value is calculated as 2pq from allele frequencies of the 1000 Genome phase 3 data, where p is the reference allele and q the alternative allele.

**Figure 3 genes-13-00122-f003:**
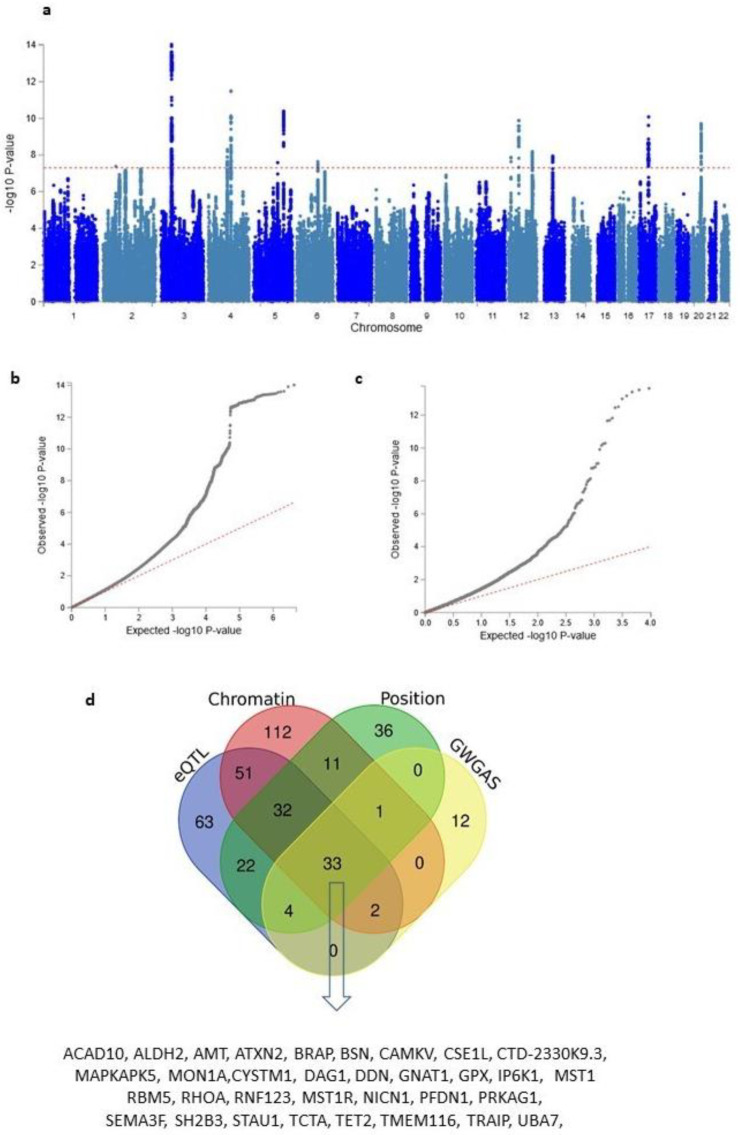
*Resilience* GWAS and gene identification. (**a**) Manhattan plot of *Resilience* identifying 13 independent genome-wide significant loci. (**b**) Quantile–quantile plot of GWAS SNPs. (**c**) Quantile –quantile plot of the gene-based association test. (**d**) Venn diagram of overlapping mapped genes by four strategies showing 33 genes were mapped by all four strategies. These genes are listed underneath.

**Figure 4 genes-13-00122-f004:**
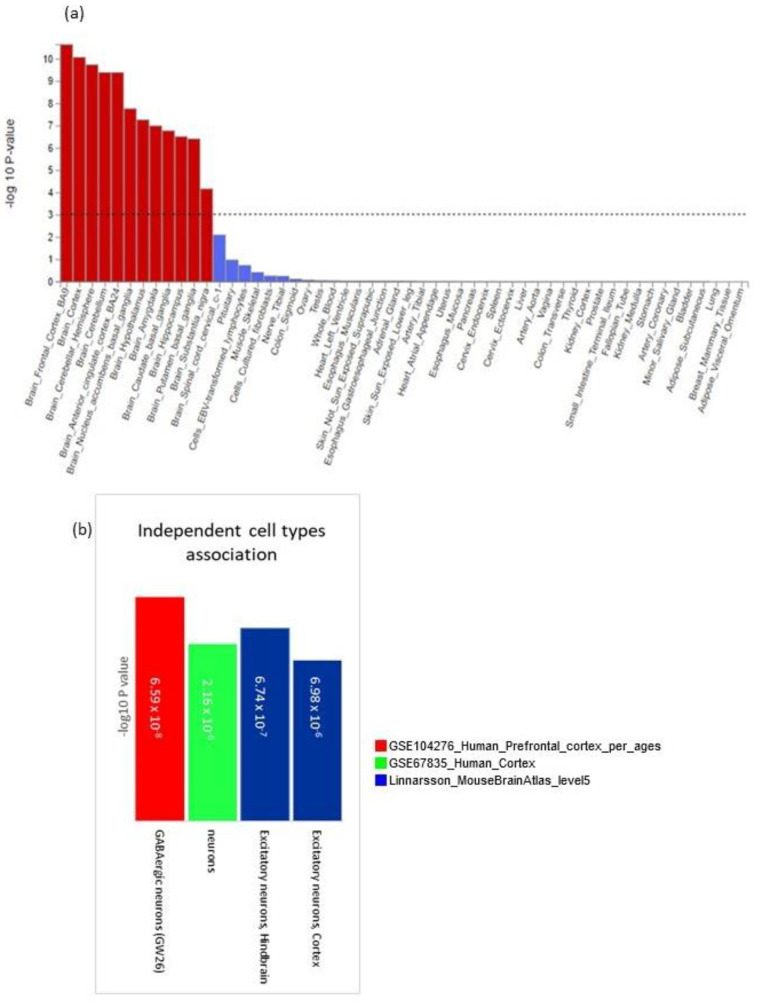
Tissue and cell type expression analysis. (**a**) Gene-tissue expression analysis based on GTEx RNA-seq data. Results that are still significant after correction for multiple testing are in red. (**b**) Independent cell type associations based on within-dataset conditional analyses.

**Figure 5 genes-13-00122-f005:**
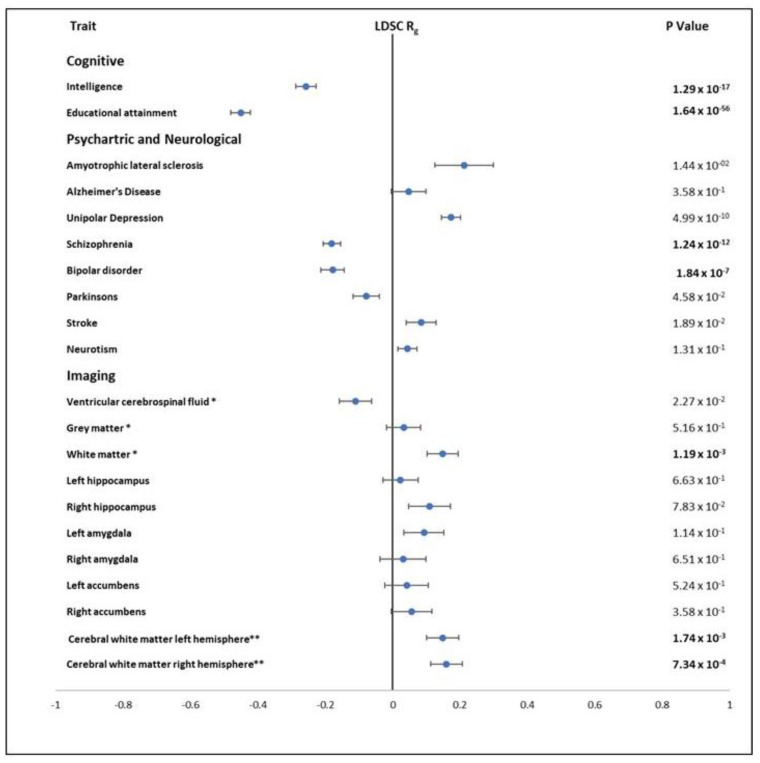
Genetic correlation with cognitive traits, psychiatric and brain disorders, and brain imaging phenotypes. Significant *p* values corrected for multiple testing are in bold, * normalised for head size, and ** generated by subcortical volumetric segmentation.

## Data Availability

Restrictions apply to the availability of these data. Data was obtained from the UK Biobank and are available at http://biobank.ndph.ox.ac.uk (accessed on 18 December 2021) with the permission of the UK Biobank. Results of functional analysis are published in Functional Mapping and Annotation (FUMA) at https://fuma.ctglab.nl/downpage.html (accessed on 18 December 2021) and coding used and *Resilience* GWAS results are available at https://github.com/joanfitz5/cog.res (accessed on 18 December 2021).

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
