# Peer review of "Thirteen Independent Genetic Loci Associated with Preserved Processing Speed in a Study of Cognitive Resilience in 330,097 Individuals in the UK Biobank"

_genes, 2022, doi:10.3390/genes13010122_

Round 1
Reviewer 1 Report
I thank the editor and the authors for giving me the opportunity to review this excellent manuscript. It presents the results from a complex in silico study of what the authors call cognitive resilience, in the context of its genetic architecture, using two excellent biobanks that have become or are becoming the gold standards for such inquiries. This is a well-executed study and well-written manuscript that presents highly original data of great interest to the field of genetics of cognitive abilities in general and cognitive resilience in particular, with profound public health implications.
It is therefore difficult for me to admit that I have strong but ambivalent feelings towards the manuscript, that I will try to outline below.
The central weakness of the manuscript lies in its phenotypic definitions. First, and most significantly, despite the overt acknowledgment of the issue, it is extremely difficult for me to accept the definition of cognitive resilience as the difference between apples (education years) and oranges (current RT some 40 years late). This is no trivial problem. Longitudinal studies of cognitive phenotypes are central to the fields of cognitive and developmental science, and even more so lately in the context of the realization of difficulties assessing meaningful change in aging populations. In fact, the concept of longitudinal validity explicitly calls out equivalence of measurement structures as the key prerequisite for inference regarding change. This concept extends to individual measures, and can (must) be explicitly evaluated using appropriate methodologies such as SEM, a method that authors relied on in a somewhat different context (more on that later). This study explicitly violates this assumption, and I am afraid I cannot accept lack of availability of relevant longitudinal measurements as a reason to proceed with the analysis that will be considered a priori flawed. I apologize for the strong language, and hope that I appropriately conveyed the gravity of the issue. Second, in their definition of the RT-based phenotype, they rely on the raw RT mean (as opposed to the median of log-transformed values in this case, but alright) -> this is a very troublesome binarization, and I am not sure it is a better approach as opposed to some sort of a discrepancy score. I discourage the latter, too, because it will not, unfortunately, save the premise of the study.
In addition to this weakness, I want to point out that I would consider diminishing processing speed, despite being a hallmark of cognitive aging, an observed manifestation of processes beyond ‘decreased neural processing speed’ as authors mention. Cognitive aging is associated with a number of neuroplastic and neurodegenerative changes that affect structural and functional characteristics of multiple networks, decreasing information processing speed, but also its efficiency and likely accuracy. It is therefore somewhat surprising to me that the authors decided to focus on RT solely, instead of enriching their study by considering gF or other, really more relevant (and proximal to the phenotypes the authors mention in the introduction, such as mild cognitive impairment) phenotypes.
Next, while I am familiar with the foundations of both GWAS and SEM, I admit I had to familiarize myself with the GenomicsSEM approach the authors used. In looking at Figure 2, was there an alternative way to conceptualize resilience using SEM without using derived measured variables that are collinear and then specifying 0 covariance between them? If alternative models were considered, can they be differentiated based on model fit, or is it prohibitively computationally expensive in a GWAS framework? The validity of inferences from this analysis depends on the validity of the specification of the model as well as definition of measured/indicator variables – the approach taken in this paper is admittedly unconventional. What was the motivation in combining discovery and replication sample under 3.3.? Unclear.
In conclusion, I cannot recommend this manuscript for publication in Genes. I am sorry I do not have better news. I do not see the way for improvement given my outlined position with respect to the key flaw of the study, its phenotypic definitions, that has great implications for findings. The findings themselves, partially replicated, are intriguing and well-contextualized using appropriate fine-mapping and prioritization approaches. Tissue-specific overrepresentation such as cortical and GABAergic prefrontal neurons is very intriguing.
Sincerely,
Sergey Kornilov
Sergey A. Kornilov, Ph.D. | Senior Research Scientist
Institute for Systems Biology (ISB) | www.isbscience.org
401 Terry Avenue North | Seattle, WA 98109-5263
P 206.732.1448 | sergey.kornilov@isbscience.org

Author Response
Reviewer 1
I thank the editor and the authors for giving me the opportunity to review this excellent manuscript. It presents the results from a complex in silico study of what the authors call cognitive resilience, in the context of its genetic architecture, using two excellent biobanks that have become or are becoming the gold standards for such inquiries. This is a well-executed study and well-written manuscript that presents highly original data of great interest to the field of genetics of cognitive abilities in general and cognitive resilience in particular, with profound public health implications.
Response: Thank you for these extremely positive comments.
It is therefore difficult for me to admit that I have strong but ambivalent feelings towards the manuscript, that I will try to outline below.
The central weakness of the manuscript lies in its phenotypic definitions. First, and most significantly, despite the overt acknowledgment of the issue, it is extremely difficult for me to accept the definition of cognitive resilience as the difference between apples (education years) and oranges (current RT some 40 years late). This is no trivial problem. Longitudinal studies of cognitive phenotypes are central to the fields of cognitive and developmental science, and even more so lately in the context of the realization of difficulties assessing meaningful change in aging populations. In fact, the concept of longitudinal validity explicitly calls out equivalence of measurement structures as the key prerequisite for inference regarding change. This concept extends to individual measures, and can (must) be explicitly evaluated using appropriate methodologies such as SEM, a method that authors relied on in a somewhat different context (more on that later). This study explicitly violates this assumption, and I am afraid I cannot accept lack of availability of relevant longitudinal measurements as a reason to proceed with the analysis that will be considered a priori flawed. I apologize for the strong language, and hope that I appropriately conveyed the gravity of the issue.
Response: We acknowledge the study limitation of the use of a proxy phenotype for past cognitive performance that is different to the measure used for current cognitive performance. However, given the pressing need in an aging demographic to understand the drivers behind cognitive decline in healthy aging and the current availability of large scale genetic and cognitive data in the UK Biobank, we explored the best way to use these data. We used the proxy phenotype of years of education (EY) to represent past cognitive performance and processing speed as measured by reaction time for current cognitive performance. Our assumption is that superior processing speed is a necessary component of academic success. Processing speed is indeed the key predictor of number sense, fluid intelligence and working memory, which in turn predict individual difference in academic achievement [1]. Studies using EEG have shown that higher processing speed can explain about 80% of variance in general intelligence. A more efficient transmission of information between frontal attention to memory storage and retrieval benefits those with higher intelligence [2]. By separating the population based on years of education, we are assuming that we are capturing individuals that in general have increased processing speed in early life. We created our final phenotype to potentially capture individuals that had preserved their processing speed over a 40-year time span.
Overall, we argue based on these data that we are not studying “apples and oranges” as the reviewer suggests. To address the reviewer’s concerns, we have updated the manuscript from line 73 to add the following: “After careful consideration and given the growing need to understand cognitive decline in an aging population we decided on an approach that would make the best use of the available data.” We had also added the following at line 83: “Our assumption is that superior processing speed is a necessary component of academic success. Processing speed is indeed the key predictor of number sense, fluid intelligence and working memory, which in turn predict individual difference in academic achievement [20]. Studies using EEG have shown that higher processing speed can explain about 80% of variance in general intelligence. A more efficient transmission of information between frontal attention to memory storage and retrieval benefits those with higher intelligence [21]. By separating the population based on years of education, we are assuming that we are capturing individuals that in general have increased processing speed in early life. We created our final phenotype to capture individuals that had preserved their processing speed over a 40-year time span.”
Second, in their definition of the RT-based phenotype, they rely on the raw RT mean (as opposed to the median of log-transformed values in this case, but alright) -> this is a very troublesome binarization, and I am not sure it is a better approach as opposed to some sort of a discrepancy score. I discourage the latter, too, because it will not, unfortunately, save the premise of the study.
Response: To clarify, we do not rely on the raw RT mean. Our RT-based phenotype is the mean of the log-transformed values of RT. It is described in the methods section as follows: ‘RT is the speed in milliseconds to correctly identify matching pairs. It was adjusted for age using the slope of the Pearson’s correlation between age and RT and for normality, using the natural log of corrected RT. A binary RT variable was created using the mean value (5.71). Those with a value less than or equal to the mean were considered to have faster than average processing speed or RT (quicker to react) and those above the mean were considered to have slower than aver-age processing speed or RT.’ This natural log adjustment was described and has been used previously in another GWAS study [3].
To address this comment, we have updated the text above manuscript to include “natural log” in line 157.
In addition to this weakness, I want to point out that I would consider diminishing processing speed, despite being a hallmark of cognitive aging, an observed manifestation of processes beyond ‘decreased neural processing speed’ as authors mention. Cognitive aging is associated with a number of neuroplastic and neurodegenerative changes that affect structural and functional characteristics of multiple networks, decreasing information processing speed, but also its efficiency and likely accuracy. It is therefore somewhat surprising to me that the authors decided to focus on RT solely, instead of enriching their study by considering gF or other, really more relevant (and proximal to the phenotypes the authors mention in the introduction, such as mild cognitive impairment) phenotypes.
Response: When selecting the cognitive phenotype to represent current cognitive performance, we examined all the available cognitive data in the UK Biobank. We are aware that the greater the power of the sample (based on sample size), the greater the likelihood of obtaining meaningful results. In addition, the parameter selected should be sensitive to changing age. To this end we examined the available base-line data within the UK Biobank and determined that reaction time was the most suitable parameter available as it had a reasonable correlation with age and was available for the most individuals. (Table 1).
Table 1: Sample size and correlation with age for different cognitive measures
|
Variable |
N |
Correlation with age (P<.01) |
|
Fluid intelligence score |
165,477 |
-0.05 |
|
Reaction Time |
496,713 |
-0.27 |
|
Numeric Memory |
51,811 |
-0.08 |
|
Visual Mem Pairs Matching |
497,926 |
-0.10 |
|
Prospective Memory |
171,569 |
-0.10 |
We also explored the use of a ‘g’ factor as suggested by the reviewer using follow up online data gathered at subsequent on-line screening. We combined the five cognitive tests performed using principal component analysis (Table 2). This “g” factor was sensitive to changing age, however a GWAS of this phenotype in the UK Biobank showed a diminished signal when compared to RT.
Table 2: Comparison of correlation with age of web-based cognitive tests (2014/2015)
|
Test |
N |
Correlation (P<.01) |
|
Trail Making (#1) Online |
104,052 |
0.27 |
|
Trail Making (#2) Online |
104,050 |
0.34 |
|
Symbol Digit Substitution Online |
118,490 |
0.43 |
|
Verbal Numerical reasoning Online |
123,665 |
0.12 |
|
Numeric Memory Online |
111,086 |
0.13 |
|
Principal Component analysis factor (g) |
111,039 |
0.39 |
To address the reviewer’s point, we have updated the manuscript at line 139 to include: “Follow-up web-based cognitive data was examined as a potential general or ‘g’ factor phenotype, however this approach had insufficient power to generate meaningful results.”
Next, while I am familiar with the foundations of both GWAS and SEM, I admit I had to familiarize myself with the GenomicsSEM approach the authors used. In looking at Figure 2, was there an alternative way to conceptualize resilience using SEM without using derived measured variables that are collinear and then specifying 0 covariance between them? If alternative models were considered, can they be differentiated based on model fit, or is it prohibitively computationally expensive in a GWAS framework? The validity of inferences from this analysis depends on the validity of the specification of the model as well as definition of measured/indicator variables – the approach taken in this paper is admittedly unconventional.
Response: We realized in our preliminary exploration that certain genetic associations were unique to samples that demonstrated resilience when compared to those that did not demonstrate resilience. However, this signal was masked by the use of the highly polygenic EY within the longitudinal phenotype. To remove this interference, we needed to subtract a GWAS of individuals that did not show resilience from those that did. We explored several manual methods to perform this subtraction, but they were unsatisfactory. GWAS-by-subtraction [4] offered a means to perform this subtraction using effect sizes and P values, resulting in a GWAS of resilience that we could then subject to functional downstream analysis. This method uses the bioinformatic tool GemonicsSEM [5] which allows for the incorporation of GWAS in structural equation modelling to perform multi variate analysis of genetic associations. The tool was described in Nature Human Behaviour in 2019 and had been cited 128 times since. It is accompanied by tutorials and user groups to assist others in its use. GWAS-by-subtraction was previously used in a study where a GWAS of cognition was subtracted from a GWAS of educational attainment to assess the non-cognitive elements of educational attainment (published in Nature Genetics in 2021 [4]). The procedure used in our study mirrors that used by Demange et al. [4], and there is a detailed description of the statistics driving this procedure in the supplementary material accompanying that paper.
We are familiar with previous research that has used SEM in a conventional way to understand the interplay of environmental, brain imaging and cognitive measures with cognitive resilience [6-9]. However, these models did not incorporate large scale genetic data. There is certainly a role for this form of analysis going forward to tease out gene – environment interactions and to study the effect of highlighted candidate genes.
What was the motivation in combining discovery and replication sample under 3.3.? Unclear.
Response: The reason we combined the discovery and replication samples was to increase the power of our GWAS before downstream functional analysis. While these samples were independent for replication purposes, they were all from the UK Biobank. It is current practice to conduct functional analysis using either a combined or meta-analysis of all available data [10].
To address this comment, we have updated the manuscript at line 387 to add: “in order to increase the statistical power of our analysis.”
In conclusion, I cannot recommend this manuscript for publication in Genes. I am sorry I do not have better news. I do not see the way for improvement given my outlined position with respect to the key flaw of the study, its phenotypic definitions, that has great implications for findings. The findings themselves, partially replicated, are intriguing and well-contextualized using appropriate fine-mapping and prioritization approaches. Tissue-specific overrepresentation such as cortical and GABAergic prefrontal neurons is very intriguing.
Response: We believe that are manuscript is scientifically sound and highlights the strengths and weaknesses of our approach.
References:
- Tikhomirova, T.; Malykh, A.; Malykh, S. Predicting Academic Achievement with Cognitive Abilities: Cross-Sectional Study across School Education. Behavioral Sciences 2020, 10, 158.
- Schubert, A.L.; Hagemann, D.; Frischkorn, G.T. Is general intelligence little more than the speed of higher-order processing? Journal of experimental psychology. General 2017, 146, 1498-1512, doi:10.1037/xge0000325.
- Davies, G.; Marioni, R.E.; Liewald, D.C.; Hill, W.D.; Hagenaars, S.P.; Harris, S.E.; Ritchie, S.J.; Luciano, M.; Fawns-Ritchie, C.; Lyall, D.; et al. Genome-wide association study of cognitive functions and educational attainment in UK Biobank (N=112 151). Mol Psychiatry 2016, 21, 758-767, doi:10.1038/mp.2016.45.
- Demange, P.A.; Malanchini, M.; Mallard, T.T.; Biroli, P.; Cox, S.R.; Grotzinger, A.D.; Tucker-Drob, E.M.; Abdellaoui, A.; Arseneault, L.; van Bergen, E.; et al. Investigating the genetic architecture of noncognitive skills using GWAS-by-subtraction. Nature Genetics 2021, 53, 35-44, doi:10.1038/s41588-020-00754-2.
- Grotzinger, A.D.; Rhemtulla, M.; de Vlaming, R.; Ritchie, S.J.; Mallard, T.T.; David, H.W.; Ip, H.F.; Marioni, R.E.; McIntosh, A.M.; Deary, I.J.; et al. Genomic structural equation modelling provides insights into the multivariate genetic architecture of complex traits. Nature Human Behaviour 2019, 3, 513-525, doi:http://dx.doi.org/10.1038/s41562-019-0566-x.
- Reed, B.R.; Dowling, M.; Tomaszewski Farias, S.; Sonnen, J.; Strauss, M.; Schneider, J.A.; Bennett, D.A.; Mungas, D. Cognitive activities during adulthood are more important than education in building reserve. Journal of the International Neuropsychological Society : JINS 2011, 17, 615-624, doi:10.1017/S1355617711000014.
- Reed, B.R.; Mungas, D.; Farias, S.T.; Harvey, D.; Beckett, L.; Widaman, K.; Hinton, L.; DeCarli, C. Measuring cognitive reserve based on the decomposition of episodic memory variance. Brain 2010, 133, 2196-2209, doi:10.1093/brain/awq154.
- Zahodne, L.B.; Manly, J.J.; Brickman, A.M.; Narkhede, A.; Griffith, E.Y.; Guzman, V.A.; Schupf, N.; Stern, Y. Is residual memory variance a valid method for quantifying cognitive reserve? A longitudinal application. Neuropsychologia 2015, 77, 260-266, doi:10.1016/j.neuropsychologia.2015.09.009.
- Marques, P.; Moreira, P.; Magalhaes, R.; Costa, P.; Santos, N.; Zihl, J.; Soares, J.; Sousa, N. The functional connectome of cognitive reserve. Hum Brain Mapp 2016, 37, 3310-3322, doi:10.1002/hbm.23242.
- Uffelmann, E.; Huang, Q.Q.; Munung, N.S.; de Vries, J.; Okada, Y.; Martin, A.R.; Martin, H.C.; Lappalainen, T.; Posthuma, D. Genome-wide association studies. Nature Reviews Methods Primers 2021, 1, 59, doi:10.1038/s43586-021-00056-9.
Reviewer 2 Report
The authors present an interesting analysis of the genetic variation associated with cognitive resilience within the UK Biobank dataset. Due to the challenges of performing controlled longitudinal assessments on complex phenotypes, such as cognitive status, the authors perform a GWAS-by-subtraction to ‘simulate’ cognitive resilience (using reaction time late in life and educational years to estimate cognitive status). This use of a ‘proxy phenotype’ for past cognitive performance is an innovative answer
to the limitation of not having direct repeated measurements of the same cognitive phenotype in a large numbers of genotyped samples over an extended time period. As such, this is represents an innovative approach to maximizing what can be gleaned from the dataset.
The goal of the analysis are ambitious and the authors acknowledge the limitations of this kind of study (esp., use of surrogate phenotypes to capture a complex phenotype (i.e., cognitive ability/resilience)). In addition, the authors are appropriately conservative in their discussion; their approach and results are provocative but both will no doubt be further refined as additional data becomes available. Also, the authors show the robustness of their method by in an independent sample. Their study presents interesting insight into how data from large cohort studies can be mined to capture complex phenotypes that will remain challenging to explicit capture in such studies.
Author Response
The authors present an interesting analysis of the genetic variation associated with cognitive resilience within the UK Biobank dataset. Due to the challenges of performing controlled longitudinal assessments on complex phenotypes, such as cognitive status, the authors perform a GWAS-by-subtraction to ‘simulate’ cognitive resilience (using reaction time late in life and educational years to estimate cognitive status). This use of a ‘proxy phenotype’ for past cognitive performance is an innovative answer to the limitation of not having direct repeated measurements of the same cognitive phenotype in a large numbers of genotyped samples over an extended time period. As such, this is represents an innovative approach to maximizing what can be gleaned from the dataset.
The goal of the analysis are ambitious and the authors acknowledge the limitations of this kind of study (esp., use of surrogate phenotypes to capture a complex phenotype (i.e., cognitive ability/resilience)). In addition, the authors are appropriately conservative in their discussion; their approach and results are provocative but both will no doubt be further refined as additional data becomes available. Also, the authors show the robustness of their method by in an independent sample. Their study presents interesting insight into how data from large cohort studies can be mined to capture complex phenotypes that will remain challenging to explicit capture in such studies.
Response: We thank this reviewer for these very positive comments.
Reviewer 3 Report
To my understanding, the Authors performed a GWAS analysis on 330,097 individuals from the UK BioBank in order to give insights into the neurobiological background of cognitive resilience. To do so, they performed a GWAS-by-subtraction analysis and, using replication samples and indipendent discovery, they find 13 indipendent genetic loci related to cognitive resilience. In particular, functional analyses showed an enrichment in cerebral tissues, for example the frontal cortex, the cortex and the cerebellar hemisphere, with no significant enrichment in other tissues of the body. Gene Ontology analysis shows also an enrichment in biological processes “neuron differentiation” and the cellular component “synaptic part”, with another significance in the “Wnt signalosome”. Mapping of GWAS results identified genes that have been previously associated with cognitive decline, synaptic activity and neurogenesis. On the other hand, a limitation of this study is represented by the Authors’ reliance on RT to create the Resilience phenotype, which results in a strong genetic correlation between Resilience and RT.
The idea presented in this Manuscript is interesting and potentially relevant to the study of the neurobiological processes that underlie cognitive resilience. The Manuscript is well written, figures are appropriate, methodology is complete and detailed and discussion is incisive.
Author Response
To my understanding, the Authors performed a GWAS analysis on 330,097 individuals from the UK BioBank in order to give insights into the neurobiological background of cognitive resilience. To do so, they performed a GWAS-by-subtraction analysis and, using replication samples and indipendent discovery, they find 13 indipendent genetic loci related to cognitive resilience. In particular, functional analyses showed an enrichment in cerebral tissues, for example the frontal cortex, the cortex and the cerebellar hemisphere, with no significant enrichment in other tissues of the body. Gene Ontology analysis shows also an enrichment in biological processes “neuron differentiation” and the cellular component “synaptic part”, with another significance in the “Wnt signalosome”. Mapping of GWAS results identified genes that have been previously associated with cognitive decline, synaptic activity and neurogenesis. On the other hand, a limitation of this study is represented by the Authors’ reliance on RT to create the Resilience phenotype, which results in a strong genetic correlation between Resilience and RT.
The idea presented in this Manuscript is interesting and potentially relevant to the study of the neurobiological processes that underlie cognitive resilience. The Manuscript is well written, figures are appropriate, methodology is complete and detailed and discussion is incisive.
Response: We thank this reviewer for these very positive comments.
Round 2
Reviewer 1 Report
I thank the authors for their detailed and thoughtful reply that also, admittedly, educated me (and triggered me to start looking into) on the GWAS SEM application as well as provided several references I was not familiar with.
Overall, I think the modifications done by the authors have improved a manuscript that I admittedly already called excellent once. It only became better.
That said, I note that the central problem I have with the study is unaddressed because it is currently unaddressable. I recognize the value in the pattern of the findings demonstrated by the authors, and in particular the tissue expression and cell type analysis are conclusive. I want to highlight that the issue of phenotypic validity is treated here in a heuristic 'best evidence available' style that for me is still just not sufficiently good. To make my position clearer, I reiterate here, the authors are using one proxy phenotype for 'past cognitive performance' that has a remarkably limited resolution ability given the number of attainable years. The problematic part (for this study) lays in the following:
Educational attainment is largely unrelated to cognitive decline (or lack of resilience) in older individuals - i.e., meta-analytic studies such as the recent study by Seblova, Berggren, and Lovden, 2020; 10.1016/j.arr.2019.101005
As a proxy variable, educational attainment is only modestly related with cognition (another meta-analysis by Opdebeeck, Martyr, & Clare, 2015; 10.1080/13825585.2015.1041450).
The availability of this phenotype may its saving grace for some, but not for me. I still feel that the paper treats this issue as not the central issue to the validity of inference and without referencing literature that is conflicting with it. I understand the historic roots of this narrative, largely enhanced by the large-scale GWASes published in high IF journals that used educational attainment as a proxy and called it 'human intelligence'. I come from the background that views intelligence as the central and known strongest contributor to academic achievement and recognizes their genetic correlations -> but it is insufficient to enable the kind of 'proxy' reasoning with construct validity implications, including for inferences like '..Over half of our genome-wide significant loci for Resilience are not associated 684 with intelligence, indicating that factors such as reserve, compensation and maintenance 685 may play a role over and above overall intelligence in determining resilience.' -> e.g, this statement is a probable direct consequence of the study design, not really a finding.
In other words, not all of us think it'a a useful phenotype, particularly when this as a limitation is mentioned in passing; and many view it as a damaging proxy phenotype (other examples include using picture vocabulary as a proxy - and that would be.. better?). Damaging because while justifying further research including into the relevance of the top candidates identified by the study, it empowers a strategy of the treatment of behavioral science phenotypes that is simply not careful enough and is neglectful, somehow, in allowing the standard of inference to suffer due to the issues related to the lack of relevant research.
I agree with the authors their research has public health relevance.
I am having a difficulty recommending this manuscript for publication.
I am also having a difficulty generating any meaningful feedback for the authors beyond thanking them again for the opportunity to get familiarized with their work. Perhaps it just does not read as realizing the matter above enough.
Author Response
I thank the authors for their detailed and thoughtful reply that also, admittedly, educated me (and triggered me to start looking into) on the GWAS SEM application as well as provided several references I was not familiar with.
Overall, I think the modifications done by the authors have improved a manuscript that I admittedly already called excellent once. It only became better.
That said, I note that the central problem I have with the study is unaddressed because it is currently unaddressable. I recognize the value in the pattern of the findings demonstrated by the authors, and in particular the tissue expression and cell type analysis are conclusive. I want to highlight that the issue of phenotypic validity is treated here in a heuristic 'best evidence available' style that for me is still just not sufficiently good. To make my position clearer, I reiterate here, the authors are using one proxy phenotype for 'past cognitive performance' that has a remarkably limited resolution ability given the number of attainable years. The problematic part (for this study) lays in the following:
Educational attainment is largely unrelated to cognitive decline (or lack of resilience) in older individuals - i.e., meta-analytic studies such as the recent study by Seblova, Berggren, and Lovden, 2020; 10.1016/j.arr.2019.101005
Response: We agree with Reviewer 1 that current finding show that educational attainment is largely unrelated to cognitive decline, however we did not use educational attainment as a proxy for cognitive resilience or decline, rather we used it as a proxy for cognition (or more specifically processing speed) in the young adult. In a subsequent publication to Seblova et al, in August 2020 Lovden et al reiterate the conclusions of Seblova et al., however they also concluded that
“the evidence indicates that educational attainment has positive effects on cognitive function. We also find evidence that cognitive abilities are associated with selection into longer durations of education and that there are common factors (e.g., parental socioeconomic resources) that affect both educational attainment and cognitive development. There is likely reciprocal interplay among these factors, and among cognitive abilities, during development. Education–cognitive ability associations are apparent across the entire adult life span and across the full range of education levels, including (to some degree) tertiary education”.
To address this point, we have updated the discussion at line 687 to add
“EY itself is not a predictor of cognitive decline [90] however it is associated with cognitive performance in younger adults [9]. Our assumption is that superior processing speed is a necessary component of academic success. Processing speed is indeed the key predictor of number sense, fluid intelligence and working memory, which in turn predict individual difference in academic achievement[91]. Studies using EEG have shown that higher processing speed can explain about 80% of variance in general intelligence. A more efficient transmission of information between frontal attention to memory storage and retrieval benefits those with higher intelligence [20]”.
(Note citation 90 is Seblova et al, as quoted by reviewer 1)
As a proxy variable, educational attainment is only modestly related with cognition (another meta-analysis by Opdebeeck, Martyr, & Clare, 2015; 10.1080/13825585.2015.1041450).
Response: While there is debate on the exact contribution of educational attainment to cognition, we refer to a 2018 review by Robert Plomin and Sophie Von Stumm ("The new genetics of intelligence") which supports the link between educational attainment and intelligence. We quote from that paper as follows:
“A breakthrough for intelligence research came from the unlikely variable of the number of years spent in full-time education, often referred to as educational attainment. Because 'years of education' is obtained as a demographic marker in nearly every GWAS, it was possible to accumulate sample sizes with the necessary power to detect very small effect sizes. Its relevance to intelligence is that years of education is highly correlated phenotypically (0.50) and genetically (0.65) with intelligence”.
To address this point, we have added the following statement to the discussion at line 700.
“There is debate within the cognitive scientific community as to the suitability of education achievement as a proxy for cognition, some showing a modest correlation [92] while others describe a high phenotypic and genetic correlation [18]”
(Note citation 92 is Opdebeeck et al, as quoted by reviewer 1)
The availability of this phenotype may its saving grace for some, but not for me. I still feel that the paper treats this issue as not the central issue to the validity of inference and without referencing literature that is conflicting with it. I understand the historic roots of this narrative, largely enhanced by the large-scale GWASes published in high IF journals that used educational attainment as a proxy and called it 'human intelligence'. I come from the background that views intelligence as the central and known strongest contributor to academic achievement and recognizes their genetic correlations -> but it is insufficient to enable the kind of 'proxy' reasoning with construct validity implications, including for inferences like '..Over half of our genome-wide significant loci for Resilience are not associated 684 with intelligence, indicating that factors such as reserve, compensation and maintenance 685 may play a role over and above overall intelligence in determining resilience.' -> e.g, this statement is a probable direct consequence of the study design, not really a finding.
In other words, not all of us think it'a a useful phenotype, particularly when this as a limitation is mentioned in passing; and many view it as a damaging proxy phenotype (other examples include using picture vocabulary as a proxy - and that would be.. better?). Damaging because while justifying further research including into the relevance of the top candidates identified by the study, it empowers a strategy of the treatment of behavioral science phenotypes that is simply not careful enough and is neglectful, somehow, in allowing the standard of inference to suffer due to the issues related to the lack of relevant research.
Response: We acknowledge the reviewers conclusion on the limitation of resilience phenotype and have added a line to the concluding paragraph at line 708 to read: “Given the limitations in the method used to construct the resilience phenotype, these findings are preliminary and will need to be confirmed by future research”
I agree with the authors their research has public health relevance.
I am having a difficulty recommending this manuscript for publication.
I am also having a difficulty generating any meaningful feedback for the authors beyond thanking them again for the opportunity to get familiarized with their work. Perhaps it just does not read as realizing the matter above enough.
Response: Thank you for giving our manuscript such thorough reviews.